# Patients’ Access to Their Psychiatric Notes: Current Policies and Practices in Sweden

**DOI:** 10.3390/ijerph18179140

**Published:** 2021-08-30

**Authors:** Annika Bärkås, Isabella Scandurra, Hanife Rexhepi, Charlotte Blease, Åsa Cajander, Maria Hägglund

**Affiliations:** 1Department of Women’s and Children’s Health, Uppsala University, SE-75237 Uppsala, Sweden; maria.hagglund@kbh.uu.se; 2School of Business, Örebro University, SE-70182 Örebro, Sweden; Isabella.Scandurra@oru.se; 3School of Informatics, Skövde University, SE-54128 Skövde, Sweden; hanife.rexhepi@his.se; 4General Medicine and Primary Care, Beth Israel Deaconess Medical Center, Boston, MA 02215, USA; cblease@bidmc.harvard.edu; 5Department of Information Technology, Uppsala University, SE-75105 Uppsala, Sweden; asa.cajander@it.uu.se

**Keywords:** mental health, psychiatry, psychiatric record, psychiatric notes, patient accessible electronic health record, PAEHR, open notes, policies

## Abstract

Patients’ access to electronic health records (EHRs) is debated worldwide, and access to psychiatry records is even more criticized. There is a nationwide service in Sweden which offers all citizens the opportunity to read their EHR, including clinical notes. This study aims to explore Swedish national and local policy regulations regarding patients’ access to their psychiatric notes and describe to what extent patients currently are offered access to them. The rationale behind the study is that current policies and current practices may differ between the 21 self-governing regions, although there is a national regulation. We gathered web-based information from policy documents and regulations from each region’s website. We also conducted key stakeholder interviews with respondents from the regions and cross-regional private care providers, using a qualitative approach. The results show that 17 of 21 regions share psychiatric notes with patients, where forensic psychiatric care was the most excluded psychiatric care setting. All private care providers reported that they mainly follow the regions’ guidelines. Our findings show that regional differences concerning sharing psychiatric notes persist, despite Swedish regulations and a national policy that stipulates equal care for everyone. The differences, however, appear to have decreased over time, and we report evidence that the regions are moving toward increased transparency for psychiatry patients.

## 1. Introduction

The use of secure web-based portals where patients can access and read their Electronic Health Record (EHR) is referred to as Patient Accessible Electronic Health Records (PAEHR). Internationally, implementation of PAEHR-services has become more widespread [1,2] but remains far from the norm. The phenomenon of sharing clinical notes or narrative visit reports with patients [3,4] is often referred to as ‘open notes’. Open notes can be considered an essential part of any PAEHR. In some countries, for example, in Sweden [2], Norway [5], Finland [6], and Estonia [7], nationwide PAEHR services, including open notes, are offered to most adult citizens. In the United States, the OpenNotes movement was initiated in 2010, providing patients access to their clinical notes [8]. Since 5 April 2021, a new federal law (21st Century Cures Act) in the US mandates all health organizations to offer patients secure online access to the information—including test results, referral information, and the notes written by clinicians—housed in their EHR [9].

However, many organizations that implement PAEHRs do not share mental health notes written by psychiatry professionals or give limited access to notes from psychiatry clinics. Research shows that the sharing of mental health notes enhances patient empowerment [2,10,11], increases the sense of control in their care [4,12,13,14,15,16] and augments patient autonomy [11,16].

Most of the studies investigating mental health patients’ experiences of access to their mental health notes have been conducted in the USA. These studies report that mental health patients experience increased understanding of their mental health [14,15], feeling in control of their care [16], and that they enhance trust in their clinician when reading their mental health notes. Further, the studies report that mental health patients experience feelings of greater engagement, validation [16,17,18,19], and that they acquire a better awareness about potential side effects of their medications when reading their mental health notes, as well as better remembering their care plan and obtaining a greater understanding of what goes on in therapy [15]. However, some patients perceive their mental health notes as inaccurate, disrespectful, judgmental, or report being surprised by disparities between what they read and what was communicated face-to-face [15,16,18,19]. Some patients also reported feeling more worried or offended by the content in their mental health notes [13,15,16,17]. In addition to deficit research being conducted on patients’ experiences on reading their mental health notes, patients suffering from severe mental illness (e.g., bipolar disorders, psychotic disorders, and personality disorders) are missing from the sample sizes. Furthermore, there has been scarce research into sharing mental health notes in inpatient or emergency care settings [20]. Studies focusing on other patient groups [21,22] or more general populations [2] have been performed in Sweden; however, psychiatric patients have to date received little attention.

Clinicians remain concerned that mental health patients may become anxious, confused or offended by what they read, and that making mental health notes accessible to patients will create more clinical work [15,23,24,25,26,27]. In a US study at the Veterans Health Administration (Washington, DC, USA), nearly 1 in 2 clinicians admitted they would be pleased if open mental health notes were discounted [24]. Similar results have been reported in a Swedish study [25]. A Norwegian study reports that 29% of clinicians in psychiatric care do not report all relevant information in the EHR when patients have access, and they keep a “shadow record” to document information they considered should be inaccessible to the patient [5]. Many psychiatry clinicians report changing their documentation due to patient access [23,24,25,26,28,29]. On the other hand, studies from the USA found that psychiatry clinicians working in outpatient settings reported greater patient engagement and perceived enhanced trust in clinicians when patients read their mental health notes [16,17].

The context of psychiatric specialist care in Sweden is viewed holistically and includes, among other things, outpatient care, inpatient care, and psychotherapy care, and often includes professionals, such as doctors, nurses, assistant nurses, psychologists, physical therapists, occupational therapists, medical secretaries, and social workers [23,25]. Therefore, the term ‘psychiatry’ will continuously be used in this paper instead of ‘mental health’.

### Sharing Psychiatric Notes in Sweden

Sweden has a decentralized healthcare system with 21 self-governing regions and private care providers spanning multiple regions. Swedish healthcare is controlled by, among other entities, the Swedish Healthcare Act (2017:30) and the Swedish Patient Act (2014:821), both highlighting the importance of care on equal terms for the entire population. The Swedish Patient Data Act (2008:355) states that the patient must have access to information about the care and treatment in order to be able to participate. Nevertheless, each region has its own policy or regulatory documents on what data patients can access in the Swedish national PAEHR service Journalen, which offers patients access to their clinical notes and see their lab results, diagnoses, referrals, medications, etc. Therefore, there are considerable differences in what health information patients have access to in Journalen, depending upon the region in which the patient has received care [2,30]. For instance, a Swedish study from 2018 reports that only 2 of the 21 regions share psychiatric notes with patients [2].

To rectify these regional policy differences, the Swedish national eHealth organization Inera, responsible for Journalen, and the Swedish Association of Local Authorities and Regions, an association consisting of the 21 regions, established the Swedish National Regulatory Framework (NRF). NRF stipulates that citizens should have direct access to all the digital health information available and the same opportunities regardless of where the citizen lives or receives care [31,32]. Inera stresses that all the regions have endorsed NRF and, therefore, that all Swedish citizens should be offered access to all health data available. Despite the self-governing regions, all the regions have agreed to deliver data from the EHRs so that all Swedish citizens’ can access their health data in Journalen. This has not been the case regarding psychiatric notes, where patient access is considered particularly controversial. In light of these concerns, this study aims to explore Swedish national and local policy regulations regarding patients’ access to their psychiatric notes and describe to what extent patients are offered access to their psychiatric notes.

This study is the first overview of how Sweden’s regions and some of the largest cross-regional private care providers share psychiatric notes. From a societal perspective, it is essential to explore current differences in Sweden, whether psychiatric notes are offered to patients or not, and why these differences occur.

## 2. Materials and Methods

In this study, we used a sequential data collection and analysis process (Figure 1). First, all web-pages from the 21 regions were analyzed, focusing on information related to psychiatry health records online. When such information was found, it was collected in an excel document to obtain an overview of the data. We studied the materials, looking for differences in how the regions provide patients with access to their psychiatric notes. We identified the following categories: (1) which region and private care provider share psychiatric notes (and for which settings), (2) if notes are shared with outpatients and inpatients, (3) if signed and unsigned notes are shared, (4) if the notes are shared with immediate access or with a delay.

Second, based on the data in the excel file, we observed differences in how the regions reported their implementation. In order to validate the gathered data and to fill out the gaps where information was missing, we performed key stakeholder email interviews with representatives of the regions (*n* = 21) and private care providers (*n* = 3). The interview questions can be found in Appendix A. The email interviews took place from the end of May to the end of June in 2021. We used a combination of structured and semi-structured questions to confirm information from the document analysis and clarify inconsistencies or missing information. Follow-up emails with additional clarificatory questions were solicited when necessary. The method was chosen to give the respondents more freedom to check the answers to the questions and the opportunity to attach documents. The answers were mapped according to the categories into an excel file, describing each region’s implementation of patients’ access to psychiatric notes. Recruitment of respondents to the key stakeholder interviews was undertaken via a closed eService used by healthcare providers who share information with patients through the PAEHR Journalen. All regions’ and the three private healthcare providers’ respective responsible administrators were reachable by the eService.

Finally, we returned to the policy and regulatory documents and performed a rapid qualitative analysis focusing on content related to patients’ access to their psychiatry notes. Relevant answers from the email interviews were also included.

The results presented in this paper are based on the qualitative document analysis, as well as the analysis of the answers from the email interviews, explaining in detail to what extent patients have access to their psychiatric notes in Sweden.

According to Swedish legislation, this study did not require ethical approval as no sensitive data were analyzed. Nonetheless, we followed ethical guidelines with informed consent.

## 3. Results

In this study, we mapped which Swedish regions and private care providers shared notes with patients in psychiatric care and conducted an analysis of *how* they are shared with patients. Additionally, we analyzed which regions currently have policies or offer regulatory documents for shared notes in psychiatric care. We also investigated which regions fully complied with the NRF.

### 3.1. Sharing Notes in Psychiatric Care

The results show that 17 of 21 regions in Sweden share notes with patients in psychiatric care (Table 1). The four regions that currently do not share notes in psychiatric care plan to start soon. All 17 regions that offer patients in psychiatric care access to their notes share notes from adult psychiatry, and 15 of these regions also share pediatric and adolescent psychiatry notes. Moreover, three regions state they plan to make notes available from pediatric and adolescent psychiatry. A representative from these regions says: “*We are looking for a secure solution in our medical record system to exclude individual conversations with children below 13 so that they are not shown to guardians*”. Another region stated that they completely exclude notes from pediatric and adolescent psychiatry.

Of the 17 regions offering shared notes in psychiatry, seven regions give patients access to notes from forensic psychiatric care. Of the ten remaining regions, two do not carry any forensic psychiatric care. Two regions plan to start sharing notes from forensic psychiatric care shortly, while two regions stated that this is currently not a priority. Four of the regions stated they decided not to share forensic psychiatric clinical notes.

The three private care providers we investigated do offer care across regions and appear to follow the regions’ guidelines on sharing psychiatric notes as far as possible. Two of the private care providers are predominantly online healthcare providers, meaning patients from all over Sweden may seek digital care from them. A representative from one of the three private care providers stated the following: “*Each region specifies if we should offer patients’ shared notes, but not with exact or detailed amounts of information. Notes from physicians, psychologists and nurses are shared on Journalen*”, while another private care provider explained they operate via one of the regions. Two private care providers are currently sharing notes in Journalen, while the third has ongoing work to start giving patients access to notes in Journalen. None of the private care providers carry forensic psychiatric care.

All the regions that share psychiatric notes share both outpatient and inpatient psychiatric notes and both signed notes (meaning, a note signed or validated by the provider who is responsible for the information in the note, indicating that the note is correct and complete) and unsigned notes (Table 2). An unsigned note is often a note that a clinician has dictated and has then been transcribed by a medical secretary and should then be checked by the clinician to confirm it is correct, a common practice in Swedish healthcare. One of the private care providers shares outpatient and inpatient notes and signed notes only. The other private care provider, which exclusively offers primary care, consequently shares signed outpatient notes only. Eleven regions and one of the private care providers share the notes with immediate access, while one of the regions only gives outpatients immediate access and 28 days’ delay to psychiatric inpatients. Four regions only share signed notes, offering immediate patient access, and unsigned notes with 14 days’ delay. Three of these regions have made it clear that they plan to make unsigned notes available immediately, while one region stated: “*Major parts of adult psychiatry routinely seal the health record established at their clinic during an ongoing care session, i.e., inpatient care. The seal is then removed in connection with the discharge. Currently, this is not going to change*”. One region and one private care provider have a delay of 14 days on all types of notes regarding psychiatric care.

### 3.2. Regional Policies and Regulatory Documents

All regions except one have a Digital Agenda or Development Strategy, where regional work on how to increase implementation and use of e-health solutions is included. In almost all regions, these documents highlight that patients should be offered access to their health records online. In the emails, the majority of the regions confirmed that they agreed to the NRF, which aims to give all Swedish citizens aged 16 or older access to all their health data in Journalen. Nevertheless, none of the regional policies identified in this study focus specifically on shared notes in psychiatry. However, from the key stakeholder email interviews, we received additional information from several regions about how they approach shared notes in psychiatry. Nine of the 21 regions report that information considered sensitive to the patient is not shown in Journalen. Thus, specific keyword templates are configured not to appear in Journalen via a form of keyword filtering. The keywords that often are mentioned from the regions are: “Early hypotheses,” “Violence in close relationships,” “Concerns about child abuse,” and “Compulsory care”. One region mentions that they have routines for how healthcare providers should document certain sensitive topics, such as suicide risk assessments or other similarly sensitive information.

An interesting discovery from the web-based search for regional documents included a regional instruction document on writing notes within psychiatry. The instructions stressed the importance of being accurate with time and event dates to be able to follow up if needed, that all people present during the visit must be documented in the notes, but that the name of the patient or relative should never be written in the notes. Clinicians are also urged not to use abbreviations or medical language, and not to use euphemisms even if they are common practice in medicine.

## 4. Discussion

This study provides the first overview of how Sweden’s regions and some of the largest cross-regional private care providers share notes from psychiatry. This study shows that all 17 regions share notes from adult psychiatry and that some regions have made a firm decision that certain psychiatric care settings are excluded in Journalen. Only one region had decided not to share notes from paediatric and adolescent psychiatry settings. The reasons behind this decision, and why it differed from other regions, are unclear.

Another interesting finding is that out of five regions that have decided not to share notes from forensic psychiatric care, one region referenced the Swedish Criminal Data Act (2018:1177) in their decisions. Only this region has interpreted that this law does not allow shared notes in forensic psychiatric care, which would be interesting to study further. The four regions and one of the private care providers that currently do not share notes from psychiatric care are planning to start soon. The results also report differences in inclusion of all psychiatric care settings (adult, forensic, paediatric, and adolescent) in the regions’ decisions to share psychiatric notes since some regions currently have decided not to share notes from forensic psychiatry and paediatric and adolescent psychiatry. Correspondingly, the results of the study report differences in access to shared notes in psychiatric care nationwide in Sweden, highlighting even more the impact of the decentralized healthcare system.

A study based on Miranda Fricker’s concept of ‘epistemic injustice’ emphasizes, among other things, denying patients access to their medical records may lead to ethical wrongs [1]. According to Fricker, the sharing and production of knowledge is a valued good; as such, inequalities in access to such knowledge and to participation in knowledge formation activities constitute an ethical wrong that can lead to primary and secondary harms [33]. In the case of shared notes, it is argued that patients who are precluded from reading their notes are thereby denied opportunities to feel more in control of their care [4,12], to better understand their mental health [15], and to facilitate patient autonomy [11,16] and empowerment [11]. Failure to access notes also means that patients cannot correct errors, omissions, or inaccuracies in their records. Blease et al. [1] argue that there is growing evidence that people with psychiatric illnesses may be more vulnerable to this type of injustice, as they are often seen negatively as unable to understand or cope with the information in their clinical notes. Should epistemic injustice indeed be an accurate portrayal of what happens when patients are denied access to their psychiatric notes, this study finds evidence of systematic structural barriers to access in Sweden. However, we also note that many, and an increasing number of, patients in Sweden do have access to their clinical notes from their psychiatric care. We emphasize that aside from the risk of ethical wrongs in denying patients to participate in their care, such as reading their clinical notes, it also violates the legal, nationally developed and agreed upon NRF. Further, more research into the practice of sharing notes is needed to confirm that any risks are minimized, for example misunderstandings due to the lack of common vocabulary between clinicians and patients [15,16,18,19].

According to Essén et al. [34], Sweden has weak legislation regarding patients’ access to their health records online, as no health data is required or mandatory by law to be shared. It also means that regions or health organizations are not fined if patients are not offered access to their health records, unlike in the USA [9]. Notwithstanding, today, 17 out of 21 regions offer patients access to their psychiatric notes in Journalen, a number that has steadily increased. Conceivably, the “soft regulation” NRF may contribute to why Sweden has this slow, yet positive, development, since a “soft regulation” will not cause any penalties if not followed [34].

### Limitations and Future Work

One limitation of our methodology was the selection of respondents for the key stakeholder interviews. We recruited participants through the closed eService used by regional and private healthcare providers when implementing Journalen nationally, with the assumption that the regions’ representatives in this group would have the knowledge to provide us answers to our questions. Most of them had a central role in working with Journalen in their region. However, they may not always have had detailed knowledge of the specific regulations regarding psychiatric notes in their respective region. If this was the case, we encouraged them to pass the questions on to the right person; however, we were not able to control whether this was necessary and if it was done.

In the study, we limited ourselves to investigating implementation policies into sharing clinical psychiatric notes rather than other data, such as laboratory results or medications. So far, most of the concerns are related to the content of the notes, and therefore we have no reason to expect stricter limitations to other types of data from psychiatric records; however, this would be interesting to explore further. We also limited the study to examining regulations regarding patients’ access to clinical notes in psychiatry at a regional level. Potentially, there may be further local, more informal practices at, e.g., the hospital, or even on individual department levels that may affect patients’ access to their psychiatric notes. In Norway, shadow records have, for example, been local practice in psychiatry that is not sanctioned in formal regulations [5]. In future research, other data collection methods could be applied to determine whether such local deviance from the policies also exist in Sweden.

We have chosen to focus on psychiatry in its entirety regarding shared notes and we have not looked specifically at access to psychiatric notes for patients who have severe mental illness. Does access to their psychiatric notes differ from patients who have mental illness that is not severe? However, since both outpatient and inpatient care settings are included in the analysis, and all regions also give access to inpatient psychiatry notes, we can assume that most patients with severe mental illness will (eventually) have access to their notes. Whether this leads to further workarounds or local deviations from the regulation similar to the Norwegian shadow records, we cannot say. Since there is a lack of research today focusing on patients who have severe mental illness and their online access to their psychiatric notes, this is, along with forensic psychiatric care, another exciting area for future work.

## 5. Conclusions

Despite the national framework NRF, which stipulates that citizens should have direct access to their entire EHR and the same opportunities regardless of where Swedish citizens live, this study enlightens that the NRF is applied differently across the regions. The enforcement is different due to the autonomous regional system in Sweden, which is observed in our results. Nonetheless, despite different enforcement, we can see that 17 of 21 regions share psychiatric notes with patients. Clinical notes from adult psychiatry are shared by 17 regions, while forensic psychiatric care is the one psychiatric care setting that is most often excluded by the regions. However, our findings demonstrate a sustained effort to implement psychiatric notes across the regions, including those that do not yet offer patients access to notes from one or more of the psychiatric care settings.

## Figures and Tables

**Figure 1 ijerph-18-09140-f001:**
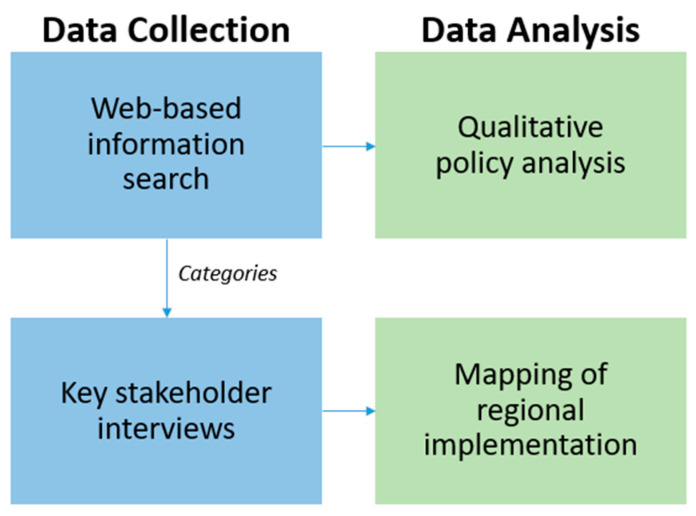
Overview of the Method.

**Table 1 ijerph-18-09140-t001:** Whether psychiatric notes are shared and, in such cases, from which psychiatric care setting for each region/private care provider. Note: (Light and dark) green colour = YES we share, (light and dark) grey colour = NO sharing, and N/A = not applicable. (Region number) 1 Blekinge, 2 Dalarna, 3 Gotland, 4 Gävleborg, 5 Halland, 6 Jämtland/Härjedalen, 7 Jönköping, 8 Kalmar, 9 Kronoberg, 10 Norrbotten, 11 Skåne, 12 Stockholm, 13 Sörmland, 14 Uppsala, 15 Värmland, 16 Västerbotten, 17 Västernorrland, 18 Västmanland, 19 Västra Götaland, 20 Örebro, 21 Östergötland. (Private care provider number) 22 Capio, 23 KRY, 24 MinDoktor.

Shared Notes in Psychiatric Care		1	2	3	4	5	6	7	8	9	10	11	12	13	14	15	16	17	18	19	20	21	22	23	24
Regions/private care providers sharing psychiatric notes	YES																								
	NO																								
Psychiatric care settings notes are available from	Adults																								
	Pediatrics–Adolescents																								
	Forensic	**N/A**		**N/A**																			**N/A**	**N/A**	**N/A**

**Table 2 ijerph-18-09140-t002:** How psychiatric notes are shared in each region/private care provider. Note: (Light and dark) blue colour = YES, and N/A = not applicable. (Region number) 1 Blekinge, 2 Dalarna, 3 Gotland, 4 Gävleborg, 5 Jönköping, 6 Kalmar, 7 Kronoberg, 8 Norrbotten, 9 Skåne, 10 Stockholm, 11 Uppsala, 12 Värmland, 13 Västernorrland, 14 Västmanland, 15 Västra Götaland, 16 Örebro, 17 Östergötland. (Private care provider number) 18 Capio, 19 KRY.

How Psychiatric Notes are Shared	1	2	3	4	5	6	7	8	9	10	11	12	13	14	15	16	17	18	19
Outpatient																			
Inpatient																			**N/A**
Signed notes																			
Unsigned notes																			
Immediate access																			
Immediate access signed notes only																			
Immediate access outpatient only																			
Respite 14 days																			
Respite 14 days unsigned notes only																			
Respite 28 days inpatient only																			

## Data Availability

Not applicable.

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
