# Peer review of "Patients’ Access to Their Psychiatric Notes: Current Policies and Practices in Sweden"

_ijerph, 2021, doi:10.3390/ijerph18179140_

Round 1
Reviewer 1 Report
A well written and well researched paper. As stated, the issue of patient access to clinical information is controversial. The authors identify the main points of contention in the context of psychiatric care and provide plausible causation for there existence.
The methodology is appropriate for this study and the use of qualitative methods enhances the meaning of the data.
I congratulate you on your paper.
Author Response
We thank you for the positive feedback!
Reviewer 2 Report
- Need more description for the qualitative research process.
- For a qualitative research of this topic, the data source based on web page and email is not enough, for example, in-depth interview with key persons.
- What's the triangulation in this study?
- How author or researcher arrange data collecting, keyword coding, concept constructing, and result framing.
Author Response
1. Need more description for the qualitative research process.
Answer: Thank you, we have revised the method description substantially and clarified the process. We have also added a figure to clarify the qualitative research method.
2. For a qualitative research of this topic, the data source based on web page and email is not enough, for example, in-depth interview with key persons.
Answer: We have also added a figure to clarify the qualitative research method, and how the data sources have contributed to the research results. In our opinion, in-depth interviews with key persons were not necessary to answer the aim of the research. A future deeper analysis would be interesting.
3. What's the triangulation in this study?
Answer: We did not do a triangulation. The methods used are now clarified in the method, and we hope that the clarity is better.
4. How author or researcher arrange data collecting, keyword coding, concept constructing, and result framing.
Answer: The methods used are now clarified in the method, and we hope that the clarity is better.
Reviewer 3 Report
This paper presents the current state in Sweden regarding the access of the patients to the clinicians' notes in the field of psychiatry.
The interesting fact is that in Sweden it seems that the system tends to reach a nationwide status for the access of patients to their psychiatric data.
Interesting onservation, but one wouls=d expect a deeper analysis and understanding of the limitations and the opportunities as well as the dangers for the healthcare system in relation with the accessibility of psychiatric data to the patients.
It would help to know the patients' view and identify the barriers and the benefits that the patients identify when using the doctors' clinical narratives.
A major issue here is whether structured data entry is used by the doctors, or free text is used. It is well known that there is a lack of a common vocabulary between physicians and patients, which may lead to problems in patient empowerment and adherence. I think the authors should address more such problems and provide metrics testing these hypothesis, so that there is more scientific depth to the paper.
Author Response
Thank you for your feedback. We completely agree about these interesting observations, and we are planning to continue to do research on patients’ access to their psychiatric notes online. We have added more relevant research to the background, clarified the background, and added further suggestions for research in the discussion.
Round 2
Reviewer 2 Report
The research process and content should be more profound for a journal paper.